# Specialised for the Swamp, Catered for in Captivity? A Cross-Institutional Evaluation of Captive Husbandry for Two Species of Lechwe

**DOI:** 10.3390/ani10101874

**Published:** 2020-10-14

**Authors:** Paul E. Rose, Lewis J. Rowden

**Affiliations:** 1Centre for Research in Animal Behaviour, College of Life & Environmental Sciences, Washington Singer Labs, University of Exeter, Perry Road, Exeter, Devon EX4 4QG, UK; 2WWT, Slimbridge Wetlands Centre, Gloucestershire GL2 7BT, UK; 3Zoological Society of London, Outer Circle, Regent’s Park, London NW1 4RY, UK; lewis.rowden@zsl.org

**Keywords:** *Kobus leche*, *Kobus megaceros*, evidence-based practice, zoo husbandry, antelope

## Abstract

**Simple Summary:**

Lechwe are social antelope adapted to wetland environments that can perform a ritualised courtship display. Although commonly housed in zoos, there is little published information available to guide their management. This study aimed to understand current husbandry practice for southern and Nile lechwe housed in North American and European institutions. A survey was sent to holders of these species, with questions addressing group demographics, enclosure characteristics, diet, enrichment, and occurrence of abnormal behaviours. Results showed that captive lechwe herds consisted of a similar ratio of male to females compared to wild herds, but there may be a limit to the number of male animals available to females at any one time. Lechwe enclosures typically featured wetland areas but these were rarely managed and there were often limited areas of vegetation for cover. The diets provided to lechwe differed when compared to existing husbandry guidelines but did not significantly differ between sampled zoos (in terms of ingredients commonly used). Abnormal behaviours were reported at several zoos but no specific causal factor was identified. This research provides a starting point for further study of the husbandry requirements of these specialised ungulates and considers the role of ecological information to the management of captive wild animals.

**Abstract:**

Lechwe are specialised wetland antelope that can have a strict social hierarchy or perform lekking during breeding. The southern lechwe *(Kobus leche)* and the Nile lechwe *(K. megaceros)* are both found in zoos globally, but little research is available to support husbandry decisions. The aim of this research was to investigate current housing and husbandry used for these lechwe across North American and European zoos. A survey was distributed to holders in 2018 and information on 33 herds (18 Nile and 15 southern) was collected. The survey focussed on population demographics, enclosure size, biologically relevant exhibit features, mixed-species holdings, nutrition, use of environmental enrichment and performance of abnormal repetitive behaviours. Results showed that lechwe were housed in herds with similar sex ratios to wild counterparts but with a potential lack of opportunity to lek. Many zoos provided wetland, but this was rarely actively managed, and not all zoos provided cover for hiding and retreat. Current feeding practice differed significantly compared to available antelope husbandry guidelines. No consistency in amounts of pellet, forage or produce provided to lechwe across institutions was found. Abnormal repetitive behaviour was noted by several zoos, but no significant predictor of such behaviour could be identified. Despite some identifiable recognition of ecology informing lechwe management, it is important that evidence-based husbandry decisions are made based on a species’ evolutionary pathway and ecological needs and some fundamental features of lechwe husbandry do not always correlate with the adaptive traits of a specialised wetland ungulate.

## 1. Introduction

For many species of wild animal housed in captivity, information on their basic care and how it compares between institutions can be lacking. Evaluation of commonly occurring practices is required before improvements or alterations can be made, based on the species’ natural history and ecological needs. Analysing what zoos do, and why, is a key foundation for the development of best practice guidelines that should promote a good quality of life in captivity for specific species. Approximately 150,000 even-toed ungulates are held in captivity across all 863 species360 © (https://www.species360.org/) member zoos (as of August 2020). Research into optimum management practices for exotic ungulates is a growing area of zoo science [1,2,3,4] and, given the diversity of populations held, and the multiple ways of exhibiting ungulates in zoos, the growth of ungulate-focussed research is useful to the development of good practice. However, not all commonly housed species are well understood.

Past research that focusses on ungulates is apparent in the literature, demonstrating how directed research into the natural history of specific species and evaluation of provision across facilities can advance husbandry and welfare standards [5]. A cross-institutional survey of common hippopotamus *(Hippopotamus amphibious)* identified numerous areas of husbandry practice and enclosure features that were not based on ecological or natural history evidence [6]; these authors noted the importance of integration of ecological knowledge into management practice to ensure that animals can experience optimal welfare conditions. The flexibility of species’ responses to environmental change also needs to be considered in zoo management practice [7]; the variation shown in responses to prevailing environmental conditions (in the wild state) provides zoos with an indication of a species behavioural flexibility and how they can adapt to the novelty presented by zoo environments.

Global husbandry surveys for specific ungulate species can identify key areas of good practice as well as deviation from any published husbandry standards [8]. Such research approaches can also form the foundation for development of husbandry guidelines, using information collected from the zoos that have been surveyed [1] to fill knowledge gaps that may be a barrier to the implementation of ecologically-sound management regimes. The focus of this research are two species of highly adapted wetland antelope; the southern lechwe *(Kobus leche)* and the Nile lechwe *(K. megaceros)*. Lechwe taxonomy is contested between two or three species, including the recently described (and Critically Endangered) Upemba lechwe, *K.l. anselli* [9]. The nominate subspecies of the southern lechwe, the red lechwe *(K.l. leche)* and the Kafue Flats subspecies *(K.l. kafuensis)* occur with relative regularity in captivity. The Nile lechwe does not subspeciate. Key ecological information for these antelope, helpful to any evaluation of captive care regimes and to provide evidence for husbandry standards, is provided in Table 1.

The populations of both southern and Nile lechwe are decreasing in the wild [10,11]; the IUCN Red List states that the southern lechwe is currently Near Threatened [10] and the Nile lechwe is Endangered [11]. As such, captive populations are relevant to population sustainability aims and potential integration into a One Plan Approach to conservation [12] would see benefits to both zoo-housed and free-living lechwe herds (in terms of population management potential). For captive lechwe herds to remain a viable conservation tool, evidence-based husbandry is required to ensure that captive care meets the ecological and evolutionary traits of the species. Such an evidence-based approach upholds the educational roles of the zoo as the animals on display are more likely to be displaying natural behaviour patterns that demonstrate their wild-type activities to the visitor.

The restricted range and small population size of the Nile lechwe [11] has led to published research on ex situ population viability to ensure long-term survival in zoos [13,14,15]. These authors note that zoos should give greater consideration to the lechwe’s ecological and behavioural adaptations when refining management style particularly breeding behaviour of lechwe, i.e., their lekking activities or hierarchical social structure during breeding for choosing a mate [16,17,18,19], when attempting to increase the genetic variability of captive populations [13] and to prevent high levels of inbreeding (that manifest in heightened infant mortality) that are caused by a lack of movement of males between captive groups of females [15]. A lek is a defined patch of ground defended by a mature male in breeding condition for the purposes of attracting females that visit these display grounds [20]. Finally, as female Nile lechwe age they are more likely to produce sons rather than daughters and the production of male calves negatively influences the future survival of the breeding female [14] due to the size of the calf at birth, and physiological and energetic costs imposed on the mother by the production of male offspring. The complex social structure of lechwe and the physiological influences on breeding success are worthy of further investigation within managed ex situ populations.

Southern and Nile lechwe are obligate floodplain grazers [21] and display the most advanced traits for a semi-aquatic existence of all species in their Order [22]. These antelopes occur in wet grasslands, being commonly found along the edges of swamps and areas of deeper water [23]. Lechwe prefer wetlands where the water depth is less than 1 m but animals will swim, leap or wade through and across deeper water to access foraging areas [10]. Zoo enclosures should consider these adaptations and habitat preferences in their design and layout. However, no species-specific guidelines for lechwe husbandry currently exist, aside from the population study conducted on Nile lechwe in 1993 [13] and information on their captive care included in an older antelope husbandry manual [24], to provide zoos with a framework to evaluate enclosure style and management regime against.

The aim of this paper is to provide information on current husbandry practices used for the two species of lechwe most frequently exhibited in European Association of Zoos and Aquaria (EAZA) and (North America) Association of Zoos and Aquariums (AZA) zoos, to enable such husbandry evidence to be used in the development of future best practice guidelines for these antelope. To date, captive husbandry has never been reviewed and published for the southern lechwe and a husbandry and population review for the Nile lechwe dates from the early 1990s [13]; therefore this survey hopes to build on older information as well as presenting a more current view to highlight the need for future research and investigation to advance evidence-based approaches.

## 2. Materials and Methods

A survey was provided to all EAZA and AZA holders of southern (red and Kafue flats) and Nile lechwe in 2018 after contact with population managers in these zoo regions. Surveys were distributed by the population managers for the species of lechwe focussed upon for this research to be filled in by a member of zoo personnel with direct responsibility for lechwe care. Twenty-six zoos provided details on 33 groups of animals (12 herds of Kafue Flats; three herds of red; and 18 herds of Nile lechwe), see Table 2 for population information. Therefore, data were acquired on 19% of global Species360-registered holders [28] for southern lechwe including subspecies (*n* = 81) and 44% for Nile lechwe (*n* = 39) as of December 2018. The survey required respondents to document number of animals housed, sizes of enclosures (including the size of indoor and outdoor areas, and details on the furnishings of the exhibits for their lechwe). Focus was given to wetland areas and the provision of ecologically relevant features within a lechwe’s exhibit. Questions within the survey were mixed; some aspects for specific counts or measurements, others were closed questions, and some were open-ended questions to enable description of specific aspects of animal husbandry to be provided. An example of counts or measurements would be respondents detailing the number of animals held (including sexes and ages), total population and intended population, as well as providing measurements of enclosures (housing, hardstanding and outdoor paddocks). Closed questions related to “yes/no” aspect of lechwe management; e.g., have “you observed aggression between your lechwe and other species, yes/no” with the option for the respondent to provide further detail if needed. Open-ended questions related to providing information on diet, enclosure features, enrichment provided and mixed species housing. Completed surveys were either returned by the postal service or via email to the second author.

Questions related to:Style of enclosure (e.g., viewed on foot or drive-through).The substrates (hardstanding, sand, dirt, wood shavings, straw, concrete) and features (rocks, branches, trees, pools, marshy areas and bushes) within the enclosure.Availability of wetland areas and how these were managed.The percentage of open to covered areas of the enclosure (i.e., due to cover and shelter provided by trees).Variety of other animal species housed with the lechwe.Description of the behaviour of the lechwe in different enclosure areas. Respondents were asked to record “yes or no” for four specific types of behaviour and then provide details on where they had ever seen these behaviour occurring in the enclosure. Respondents reported where they had seen their lechwe (pool/wetland, paddock, under cover) performing key state (long-duration) behaviours (rumination, feeding, socialising, resting/lying down).Descriptions of any negative social interactions (e.g., chasing, fighting, overt aggression and dominance) with the lechwe and other animal species (if applicable). Again, these were self-reported by each individual zoo.Description of any reported abnormal repetitive behaviour, ARBs, (“stereotypic behaviour”) in the current lechwe herd at the respondent’s zoo. Respondents were given instructions that ARB related to any repetitive, invariant, stereotypic behaviour; for example, pacing, head-rolling, oral tongue-playing behaviours.

Respondents were also asked to provide details on the diet fed to the lechwe, including amounts and brands fed, as well as the number of feeds per day and the ratio of concentrate pellets, forage and fresh produce that made up a daily diet (on a herd or individual basis). Questions on supplements given, use of browse and seasonality of feeding style were also included.

Finally, respondents were questioned on the use of environmental enrichment for the lechwe, the social environment provided for the lechwe at the respondent’s institution and whether the herd of antelope was considered as an enrichment factor for management.

For the purposes of analysis, the subspecies of southern lechwe housed in this sample of zoos (*K.l. leche* and *K.l. kafuensis*) have been combined due to their ecological similarity (O’Shaughnessy 2010; Schuster 1976). Again, due to similar ecological overlap [18,29], habitat preferences and behaviour [30] published guidelines on diets for zoo housed kob (*K. kob*) were used as a baseline for analysis with southern lechwe diet [24] as no lechwe-specific information could be found. Evaluation of the diet information provided by this sample of zoos for their Nile lechwe could be directly compared to the published guidelines [24].

### 2.1. Data Analysis

Data were analysed in R v3.6.1 (R Foundation for Statistical Computing, Vienna, Austria) [31] using R studio v. 1.2.1335 (R Foundation for Statistical Computing, Vienna, Austria) [32]. Anderson–Darling tests were used to determine the normal distribution for specific subsets of data from within all data from all surveys. For all regression or fitted models, the “plot(name of model)” function was used in RStudio to check the distribution of residuals and predicted values to see how randomly distributed points are, and the normal Q-Q plot to see how much error deviates from normality. No transformation of data occurred to create normally distributed data points.

#### 2.1.1. Housing and Enclosure Features

Areas of lechwe paddocks, hardstanding and indoor housing provided in survey responses were not all normally distributed and Spearman’s rho correlations were applied to determine any relationship between measurements. To identify any significant predictors of paddock size provided to the lechwe in this sample, that might be suggestive of standardised enclosure design across zoos, a general linear model was run in RStudio and the output presented from the “anova (name of model)” function as well individual variable estimates, t values and *p* values tabulated. The output variable was the size (area) of the outdoor paddock and the predictor variables were: the lechwe species, the number of years kept, the association that the zoo was in (EAZA or AZA), the total herd size, the intended herd size, whether the enclosure was drive-through or not, the degree of cover provided in the enclosure, the presence of a wetland, whether the institution was breeding the lechwe or not, and whether the enclosure was mixed-species. Where no information was provided on the intended population, the total population currently housed was used as the inferred intended population.

Post-hoc testing of categorical predictors that approached significance were investigated using the “lsmeans” [33] and “pbkrtest” [34] packages in RStudio, for continuous predictors, model estimates were evaluated to determine the direction of any effect. For continuous predictors that approached significance, scatterplots were drawn to visually assess the trend between output and predictor variables. Confidence intervals, calculated from least squares means post hoc testing, are quoted for categorical variables.

#### 2.1.2. Nutrition

Data on forage percentage and pellet percentage per captive diet were tested for normality and found to be normally distributed. To evaluate any difference between the proportion of lechwe diet made up of pellet and of forage compared to the 1999 AZA published guidelines for Reduncinae antelopes [24], a one-sample t-test was run and corresponding intervals plots are presented. A one-way ANOVA was run on the proportion of pellet and of forage provided at each zoo to compare between species.

Nine lechwe herds (five southern, four Nile) were provided with produce as part of their daily ration. For these nine herds, produce ration was also normally distributed and so a two-sampled t-test was run to see if there was any difference in amount of produce provided to each species.

#### 2.1.3. Behaviour, Environmental Enrichment and Ecologically Relevant Resources

For those institutions that provided a wetland (to ensure that fair representation of occurrence of behaviour would be included), a basic time-activity budget was created by counting the self-reported occurrences of state behaviours (rumination, foraging, resting and socialising) of lechwe from each responding institution and dividing by the total counts of all behaviour noted across respondents. These data were then compared to published time-activity budgets for male and female red lechwe by Williamson [35] and by Lent [36], who counted occurrences of active and inactive animals in the Okavango during December and January. Percentage occurrence of active animals was averaged from Lent’s daytime data (from 09:00 to 17:00) based on when zookeepers would be around to determine activity patterns of the animals sampled in this survey. A record of where lechwe were reported as performing key state behaviours were taken from each survey with pool and swamp features combined together. A cross tabulation and Chi-squared test was run on the percentage of observations of active (foraging, moving, socialising) and inactive (resting and ruminating) lechwe in the survey and in published information on wild animals to see if there was any difference in frequency of activity in captive herds.

Data on reported behaviours and where they were seen were counts and therefore a Poisson regression was run on the records of behaviour (social, forage, rest and rumination) from each survey for three “habitat types” (grassland, wetland, cover) to determine any difference in the observation of behaviour in each of these “habitats”.

As wild lechwe are documented to rest near water [37], and to analyse any influence of husbandry and environmental predictors of observation (yes or no) of resting and of rumination in wetland areas (pool and swamp combined), a binomial regression [using the glm function in RStudio, family = binomial (link = ”logit”)] was run. The size of the wetland was categorised (based on the number of responses and the range of areas detailed) into small (30 to 199 m^2^), medium (200 to 450 m^2^), large (500 m^2^ +) and unknown. Predictors included in the model were: herd size, wetland size, paddock size, space per animal and lechwe species.

The same binomial GLM with logistic link function was used to understand any predictors of observation of stereotypic behaviour (yes or no); with occurrence of stereotypic behaviour as the outcome variable and paddock size, house size, hardstand size, provision of browse, number of feeds per day, whether a drive-through enclosure or not, presence of a wetland, intended herd size, number of females in the herd, species of lechwe, how open the enclosure was (as a percentage) and whether the enclosure was a mixed-species exhibit were included as predictors. The “rsq” package was used to generate r2 values to check the amount of variation captured by these binomial GLMs, and (for all models) the “plot (model name)” function was used to check residuals and normal Q-Q values for model fit.

Three zoos reported that no area of the paddock was covered yet all three reported the presence of living trees in the enclosure, which would provide cover. Consequently, Google Maps (Google LLC, Mountain View, CA, USA) was used to estimate the degree of tree cover of the paddock for these zoos.

## 3. Results

### 3.1. Housing and Enclosure Features

Key information on the size of the enclosure provided and the number of animals housed within, as well as the provision of wetland for the lechwe is summarised, for each institution, in Table 3.

Southern lechwe were more commonly reported on from the EAZA region compared to from the AZA region, but across all institutions, more female animals are housed compared to the number of males (Figure 1). A bias in European holding of southern lechwe (and a small number of males to females) is reflected in current data that show 926 southern lechwe in the European region and 89 in the North American region [28].

The maximum amount of wetland per animal was 635 m^2^ per animal (paddock size 29,149 m^2^, herd size 10 animals) and the smallest (of those zoos that provided wetland) was 4 m^2^ per animal (16,500 m^2^, herd size 13 animals). The median amount of wetland per animal (*n* = 17) was 28.6 m^2^ per animal.

Spearman’s rho (rs) correlations show no relationship between house area and paddock area (rs = −0.26, *p* = 0.152), paddock area and hardstand area (rs = −0.03; *p* = 0.867) or for hardstand area and house area (rs = 0.16; *p* = 0.374). When modelling relevant potential predictors of paddock size, there are significant predictors of area provided to lechwe from this dataset, F_10,22_ = 6.382; r^2^ = 63%; *p* < 0.001, with model output provided in Table 4. *p* values have been compared to a corrected alpha level of 0.01 [38] and significant Q values highlighted using * and in italics.

There is a highly significant relationship between an increasing number of lechwe housed and a greater area of outdoor paddock (Q = 0.005) as well as a significant effect of overall intended herd size on the area of the animal’s outside paddock (Q = 0.01). There is a non-significant trend (Q = 0.015) for larger paddocks to contain less cover. The area of an outdoor paddock provided to a non-breeding group of lechwe was larger (Confidence Intervals, CI (16,662, 90,142)) compared to that provided to a breeding group (CI (7087, 54,725)) although this relationship only approached significance (Q = 0.02). For lechwe in drive-through enclosures, paddock area was generally larger (CI (16,658, 81,568)) compared to enclosures that were not drive-through (CI (5936, 64,455)) but this relationship was not significant (Q = 0.04). No significant difference is apparent for paddock size provided to southern (CI (4067, 65,197)) or Nile (CI (18,034, 81,319)) lechwe (Q = 0.03).

#### Mixed-Species Exhibit (MSE) Details

Twenty-seven (82%) of the 33 lechwe herds surveyed were housed in mixed species exhibits; 16 (89%) of Nile lechwe herds and 11 (73%) of southern lechwe herds. For all lechwe in mixed species exhibits, other species were always Artiodactyla (e.g., giraffe, *Giraffa camelopardalis*), Perissodactyla (e.g., zebra, *Quagga* sp.) or birds (e.g., ostrich, *Struthio camelus*). A summary of these species mixes is shown in Table A1 of Appendix A. Of those lechwe housed in MSE, 50% of Nile and 55% of southern lechwe (52% across both species overall) herds reported instances of interspecific aggression, e.g., chasing, fighting, displays of dominance and all of these aggressive interactions were performed by male lechwe (for both lechwe species). Aggressive interactions between lechwe and other named species are detailed in Table A2.

### 3.2. Nutrition

For southern lechwe, there was a significant difference between the amount of forage, e.g., hay or similar, provided by these sample zoos (t = 5.63; μ = 72.8%; *n* = 12; *p* < 0.001) and the amount of forage recommended for kob (50%) in the 1999 AZA published guidelines. The same significant difference was noted from these sample zoos (t = 6.16; μ = 78.1%; *n* = 11; *p* < 0.001) and for recommended forage (43%) for Nile lechwe. There was a significant difference for pellet (57%) proportion too (t = −8.96; μ = 15.27; *n* = 11; *p* < 0.001) for the Nile lechwe. Likewise, for southern lechwe there was a significant difference between the suggested proportion of pellet (50%) in 1999 AZA published guidelines and the amount provided in these sample zoos (t = −7.93; μ = 21.5%; *n* = 12; *p* < 0.001).

Figure 2 compared the average across forage, pellet and produce in the diets for these surveyed Nile and southern lechwe against the recommended guidelines of forage and pellet from the 1999 AZA guidelines [24]. Across all sampled zoos, there was no significant difference in the amount of forage (F = 0.58; df = 1, 21; *p* = 0.455) or pellet (F = 1.14; df = 1, 21; *p* = 0.297) provided to either southern or Nile lechwe. For the nine herds of lechwe provided with produce there was no significant difference in the amount of produce given to either species (t = 0.557; Nile μ = 18.3; red μ = 13.6; *n* = 9; *p* = 0.595) even though Nile lechwe are provided with more produce in their diet compared to southern lechwe. Thirteen zoos (16 herds) provided some form of seasonality to dietary provision, and this may include seasonal variation in use of browse for nutritional enrichment (see Section 3.3).

### 3.3. Natural Behaviour, Environmental Enrichment and Ecologically Relevant Resources

Of the total 33 herds of lechwe, 23 herds from 18 institutions were provided with a wetland within their enclosure. Active management of a wetland within the enclosure was only conducted by one institution, suggesting that the wetlands within these lechwe enclosures were naturally occurring or areas of the exhibit prone to flooding. Of these 23 herds, data on wetland size was provided for 17 herds (14 institutions). Eleven herds of southern lechwe were provided with a wetland (for those with data, 86.45 m^2^/animal) and 12 herds of Nile lechwe (for those with data, 113.3 m^2^/animal). For these 17 herds, 100% were mixed species exhibits and 58% were breeding herds, and 71% displayed no stereotypic behaviour.

Twenty-four zoos provided details on where their animals performed key state behaviours, which in turn provided information on the behaviour patterns of 31 lechwe herds (15 red and 16 Nile). Out of these records, all lechwe herds were noted as socialising in open grassland and all but one herds were noted as foraging, resting and ruminating in grassland areas too (Figure 3). Further evaluation of these behavioural data was undertaken for the subsample of 16 zoos (21 herds, 11 southern and 10 Nile) that were provided with all specific enclosure zones noted in the survey (wetland, grassland and cover).

Data on records of behaviour in each enclosure zone were normally distributed (*p* = 0.177). Counts of each behaviour from Figure 3 were inputted into a Poisson regression in RStudio with “habitat” (i.e., enclosure zone) and behaviour as the predictor variables. Post-hoc testing found no significant difference between records of each behaviour but did find significantly more records of behaviour noted in grassland, compared to wetland (estimate = 0.412, SE = 0.174, Z ratio = 2.367, *p* = 0.047). The r^2^ for this model was 90%.

For the 21 herds provided with a wetland, a logistic regression was run to identify whether resting and rumination occurred preferentially in wetland areas. There are no significant predictors of wetland usage for either behaviour (Table 5), the r^2^ value for the rumination model is 55% and for the resting model, r^2^ = 41%.

Rumination frequency from the survey data is similar to that observed in wild animals but socialising is much lower in the wild compared to that noted in surveys (Figure 4). There is no difference between records of activity or inactivity for captive and wild lechwe. Cross tabulation Chi-squared testing identifies no significant difference between inactive and active observations on captive or wild lechwe (χ^2^ = 0.721; df = 1; *p* = 0.396).

Across all surveys, 24% of respondents documented ARB performance in their animals. All reported ARBs were pacing behaviours, with specific details relating to pacing along a fence line in female lechwe immediately before parturition. A binomial GLM with logistic link (Table 6) shows there to be no significant predictor of occurrence (yes/no) of stereotypic behaviour in this sample of lechwe. The r^2^ value for this model was 47%.

Twenty-seven responses considered the overall herd as a form of social environment (21 zoos), and nine responses (8 zoos) stated they provided extra enrichment to their animals. Twelve zoos (15 herds) stated they provided extra browse to their lechwe, excluding that which may be naturally growing within the enclosure. Thirty herds had access to living trees in their enclosure and 16 herds were provided with bushes and shrubs within the enclosure too. Only two zoos responded saying that lechwe had no grass in their outside paddock, being maintained solely on a sand paddock, and 14 herds (11 zoos) were provided with areas of long grass within their enclosure. Thirty herds (23 zoos) were provided with soil substrate outside and 26 herds (20 zoos) had a sanded area as part of their outdoor paddock too.

## 4. Discussion

This husbandry survey provides useful information on the management practices experienced by 33 herds of lechwe (18 Nile and 15 southern, 3 red and 12 Kafue Flats). Overall, our findings highlight variation across EAZA and AZA institutions in the style of management utilised for these antelopes. Some standardised management practices were identified. These included: grassland within outdoor paddocks, diets formed of forage and concentrate pellet, and social housing in mixed sex groups with more females to males, which mimics data on wild social structure [39]. Inconsistent practice was noted for enclosure size and space, as well as the features provided within the enclosure, see Table 3, and for the amounts of components of each diet fed to herds at each institution. Space provided to lechwe in zoos was significantly influenced by the zoo’s current herd size as well as the projected herd size. This is to be commended, particularly in the case of the Nile lechwe where the aggressive behaviour of males towards one another to establish a social hierarchy and the reclusive behaviour of pre-parturient females [13,19], is suggestive of a need for as much space as can be provided (i.e., larger spaces are better for natural social structures and hence for lechwe welfare).

Whether the zoo is breeding the lechwe and the degree of cover (from vegetation) provided also shows a general trend with outdoor paddock area, suggesting that as enclosures get larger, they can become more uniform, suggesting that quantity of space is considered more in the enclosure design for these antelopes than habitat features or structure. Whilst not significant, lechwe behaviour shows variation between different habitat features provided within an enclosure (Figure 3) and as a proportion of observations, variation of observed behaviour was highest in wetlands and in areas of cover with a more uniform observation of key activities noted in grassland. This may reflect the predominance of grassland in lechwe areas for these sampled zoos or the removal of predation risk in zoo-housed environments, which encourages the antelope to graze more out in the open. The crude way of assessing animal behaviour from these survey questions does not provide information on how long animals spend on each behaviour in each zone, nor does it tell us where all individuals perform important state behaviours that link to positive welfare, i.e., rumination. However, Figure 3 provides important information for guiding future research; assessment of time-activity patterns within specific zones to determine how behavioural diversity in the zoo compares to published information from the wild. This would further provide support for our basic evaluation of how naturalistic the activity of these lechwe are (Figure 4) and could provide the basis for welfare assessment that used degree of inactivity as an animal-based welfare score.

Increasing the size of wetland spaces and cover may be beneficial; 85% of zoos reported socialisation occurred in wetlands for example (Figure 3) and this figure may be higher if all zoos in the sample either: (i) provided information on the size of their wetland; or (ii) provided a wetland to their lechwe. Due to the simplicity of its collection, behavioural data gathered by this survey may be lacking in its ability to pinpoint the overall importance of wetland areas to the physical and psychological health of these lechwe. We encourage further research into space use and performance of key behavioural indicators of welfare in captive lechwe, as has been demonstrated by research on captive sitatunga (*Tragelaphus spekii*) [40] and specific methods of defining resources within an enclosure and assessing their value to the animal are available in the literature [41,42]. Such an approach would help build on the foundation of “what lechwe are seen to do and where” that are presented here to further inform species-specific best practice guidelines that have been shown to be relevant to enhanced species management in other specialised ungulates, such as Eld’s deer *(Panolia eldii)* and lesser kudu (*T. imberbis*) [43].

Zoos should also consider expanding on the wetland areas of their lechwe exhibits to promote the performance of behaviours that would naturally occur in such areas, which would be beneficial to animal welfare and to visitor education. Male lechwe can be 145% of an average adult female weight [44] and this sexual dimorphism is important for territorial display and courtship behaviour. Ecological separation of male and female lechwe in foraging patches may also be due to this size difference [37] and small individuals can be pushed out of profitable feeding areas. This marked size difference should be considered by zoos when designing lechwe enclosures so that any ritualised chasing or display between males can be performed without harassment or disturbance to females and young, which is noted as occurring in wild herds [25].

For both species of lechwe studied, formulation of each zoo’s lechwe diet did not match, for either species, the amounts recommended in the Kendall and Rieches [24] husbandry manual (Figure 2). The differences in real-world feeding practice may be explained by the age of the guidelines and the evolution of zoo dietary knowledge, where more forage is beneficial to captive ungulate health alongside of a restricted, measured amount of concentrate pellets [45,46]. A review of zoo health records to assess condition of animals on different types of diet (different proportions of forage to pellet, with and without produce) would provide evidence for the most suitable diet formulation for lechwe in different zoo regions (considering climate and weather variables on animal metabolism and homeostatic demand).

Wild lechwe are noted to be seasonal in their diet selection and select for a wide range of different types of vegetation across a large area [44]. This variation in feeding ecology should guide zoo feeding practice via the increased use of browse as well as access to grazing in all cases. Hofmann [47] categorises close relatives of the lechwe, the kob and the waterbuck (*Kobus ellipsiprymnus*) as grazers rather than intermediate feeders or browsers, so whilst only 12 zoos provided browse to their lechwe, this may well be a relatively high proportion when compared to the ecological need for browse in this species. However, as a form of enrichment and to promote foraging and rumination, we recommend all lechwe holders consider browse as a forage option. Thirteen zoos attempted some form of seasonal variation in how diet and browse was provided, which aligns with the natural ecology of lechwe showing seasonal choice in selection of vegetation when grazing. Further assessment of intake of captive diets on an individual animal basis, as well as proximate analysis of diet content would help evaluate the standard of feeding for captive lechwe. Data on wild plant selectivity is available [44,48] and hence information on natural feeding patterns could be incorporated into zoo diet formulation. Degree of foraging within an enclosure, which would augment individual daily intake, is difficult to measure accurately, but how much ad hoc feeding is conducted by lechwe in naturalistic enclosures compared to those managed on sand yards (with restricted browsing and grazing opportunities) could be assessed alongside physical condition (e.g., coat, tooth and hoof health) as well as general activity to evaluate the effect of feeding regime on animal welfare.

The institutions sampled in this survey mixed their lechwe herds with a wide range of species, with most popular mixes with other mammals being giraffe (16 records), zebra (11 records), kudu (10 records) and eland (9 records). The most common non-mammal housed with lechwe was ostrich (8 records). These mixes show ecological overlap that could naturally occur for wild southern lechwe herds and the populations of these other species, but may not be as ecologically relevant for Nile lechwe. Further investigation of how Nile lechwe utilise an enclosure when housed with other species in captivity is needed to provide evidence for the most appropriate mixes of different species. Such an approach has worked for sitatunga, another wetland specialist, and individual animal observation provides useful data on the importance of aquatic resources within zoo environments [40].

Wild lechwe clearly show a preference for shallow water foraging and are specifically adapted for the consumption of wetland grasses [49]. Whilst complete replication of such a habitat could be logistically challenging in captivity, zoos should move away from completely sanded outdoor paddocks, which may offer little in the way of environmental enrichment, towards the creation of more heterogenous environments that mimic key facets of the lechwe’s ecosystem. Housing lechwe in mixed species enclosures with other ungulate species that come from drier grasslands may be limiting the degree of swamp-like features in their zoo exhibits. Further investigation into any health impacts of maintaining lechwe on homogenous, dry paddocks should be carried out to supplement investigations on a behavioural basis.

Zoos may need to consider the size of lechwe enclosures to enable leks to form, if multiple males are kept. Based on the life history strategy of these species, multiple males should be kept to enable genetic diversity to be maintained (and males need to be moved around groups accordingly). Size of leks corresponds to the number of breeding males in an area, i.e., the higher the number of males, the higher the likelihood of leks being formed [26]. Red lechwe and its subspecies congregate in herds of single sexes outside of the breeding season [16] and future zoo enclosure design could consider this seasonal social structure in terms of facilities (e.g., holding areas and paddock separation) for captive lechwe herds. Integration of mate choice can help improve conservation breeding outcomes [50] and given the importance of mate choice to lechwe reproduction, more investigation into multi-male, multi-female groups and the interactions between animals could be useful to the development of species-specific management guidelines.

Wetland areas are important to adult lechwe activity patterns, with research showing that, in the dry season, female lechwe will spend 42.3% of their time in water, adult males 28%, and subadults 2% of their time [51]. Nile lechwe are recorded as spending almost all of their time in shallow water [48]. Therefore, zoos that do not provide wetland may be depriving individual lechwe of a key ecological requirement and this should be reconsidered in light of animal welfare needs and species-appropriate husbandry. Further study that investigates the time spent in specific enclosure areas (e.g., wetland) and the diversity of behaviours performed in such enclosure areas would provide valuable evidence for the importance of access to water for captive lechwe.

Male lechwe rest during the daytime for longer periods of time compared to females [52]; high records of inactivity noted in the survey responses (Figure 4) may be indicative of keepers being most knowledgeable of their animals’ behaviour during normal daytime working hours. Nocturnal activity needs to be investigated as grazing occurs at night [36] even though overnight behaviour is poorly documented [30], and hence the welfare of zoo-housed individuals may be compromised if forage is not accessible in overnight housing. The large proportion of behaviours observed in the grassland maybe self-evident because grassland is the largest area of the outdoor space provided to these lechwe herds at these facilities. Further research that analyses zone size compared to the other areas that might be more ecologically useful (e.g., wetland) would help determine the lechwe’s preferences for enclosure occupancy.

Eleven of these sampled zoos provided areas of long grass for their lechwe. We encourage all zoos to consider planting and grassland management techniques to provide such habitat features within the enclosures of all lechwe. Wild lechwe change their reliance on water based on physiological state and life stage. Female lechwe with young are more likely to remain near to water as they are warier of their surroundings, and both sexes of lechwe retreat into long grass and thick reeds when disturbed by predators, but females with calves are more likely to venture into thick vegetation compared to males [36]. Measurement of lechwe welfare in captivity could be based on wetland usage and around performance of seasonal variation in time-activity patterns, which is well documented for wild animals [35,36,51]. Degree of time spent standing and lying “idle” is recorded for wild herds of different sexes and ages [35]; consequently if wild lechwe are known to spend a proportion of their day in a general inactive state, measurement of time spent inactive in zoo housed specimens could be compared against zoo enclosure features (i.e., a lack of suitable areas for socialising or foraging), herd structure and feeding regime to assess naturalistic behaviour patterns. Heightened levels of inactivity could be used as evidence of a need to alter enclosure style or animal care practice.

Both species of lechwe included in this project are of conservation concern [10,11] and some note that the future population survival of the Nile lechwe to be “precarious” in some parts of its range [48] due to anthropogenic threats. Consequently, the captive population of lechwe is of conservation value, particularly if such herds are ever used in a “One Plan Approach” to conservation [12] that sees captive individuals being used for direct conservation action for wild herds (for example as a source for future augmentation of wild herds or reestablishment of wild herds if needed). This should encourage zoos to further scrutinise lechwe husbandry practice to ensure that both individual needs across life stage as well as population sustainability and viability are catered for. Nile lechwe have, in the past, been documented as faring poorly in captivity due to an inability to adapt to human presence [13] and, in a study of 12 ungulate species’ behavioural responses to human presence Nile lechwe were one of two species that displayed stereotypic pacing behaviour [53], suggesting that this species may be more challenging to maintain comfortably in captive settings. Whilst not tested as part of this research (due the lack of significant predictors of ARB from these data, Table 6) and the small number of reports overall, more ARBs was observed in herds of Nile lechwe (five records) compared to southern lechwe (three records), therefore providing support for further investigation of the behavioural responses of this species to captivity.

Finally, as the social dynamic of a Nile lechwe herd is yet to be fully defined [14] it may be that gaps in our knowledge of the behavioural ecology of this species are limiting the development of husbandry guidelines. Therefore, further evaluation of information on the wild ecology of this species is required as well as more detailed investigation of captive animals and their responses to a managed environment to further provide evidence for best practice of this lechwe species. The relatively small number of male southern lechwe housed across AZA and EAZA zoos may need further scrutiny if population viability is to be maintained in a species where its specialised system of mate choice means only a dominant male is chosen by breeding females. Previous work has demonstrated how quickly inbreeding and associated depressive effects on genetic quality can appear in captive lechwe herds [13,15], therefore zoos should work together to encourage more holders of both species to provide a gene pool of individuals suitable for the maintenance of genetic variation and key adaptive traits.

## 5. Conclusions

This survey has identified numerous differences in lechwe management, for both species, across these sampled zoos. Whilst all zoos managed their lechwe in herds, space provided to captive lechwe was highly variable. Zoos all fed the same dietary format of pellets, forage and some produce and this was similar for both species, variation in the amount of all three dietary ingredients is noted. Eighty percent of these zoos considered the herd a lechwe exists within to be its main form of environmental enrichment and managing this species is a social group may account for the low responses on noted performance of ARBs (24% of responses). Zoos within this survey are aware of the lechwe’s specialised adaptations to a wetland environment, but certain aspects of lechwe husbandry appear to lack an evidence basis. These lechwe were mixed with a range of different species of animal; some that have an ecological validity, others that may share similar habitat needs, and some that have no ecological similarity to the lechwe. We recommend further study in the enclosure usage of these antelopes, their behaviour and social organisation, to determine the most relevant herd size and structure, and paddock size, for sustainable breeding, the performance of natural behaviour patterns and attainment of good animal welfare.

## Figures and Tables

**Figure 1 animals-10-01874-f001:**
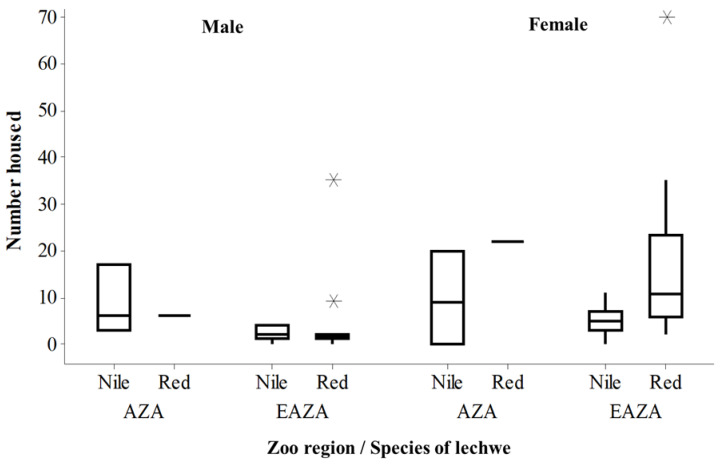
Comparison of number of male and female individuals in herds across EAZA and AZA regions for both Nile and southern (“red”) lechwe. Boxplot showing the median and interquartile ranges, as well as outliers (marked as asterisks).

**Figure 2 animals-10-01874-f002:**
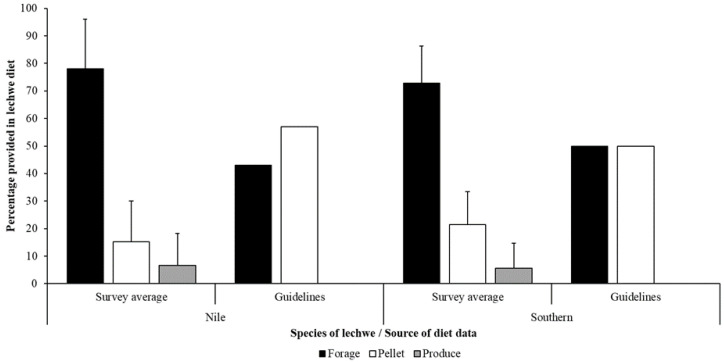
The average (mean plus standard error) for all data on proportion of forage (black), pellets (white) and produce (grey) for Nile (left) and southern (right) lechwe in the surveyed population compared to the suggested proportion of forage and pellets for *Kobus* antelope species available in the 1999 AZA published guidelines.

**Figure 3 animals-10-01874-f003:**
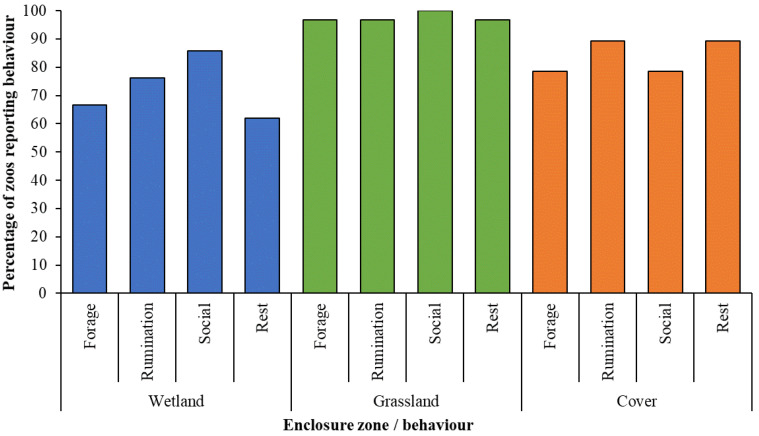
Self-reported observation of key state behaviours performed per enclosure zone from all surveys that noted lechwe behaviour. Percentages are calculated from records of behaviour from those zoos that stated their lechwe were provided with that enclosure zone.

**Figure 4 animals-10-01874-f004:**
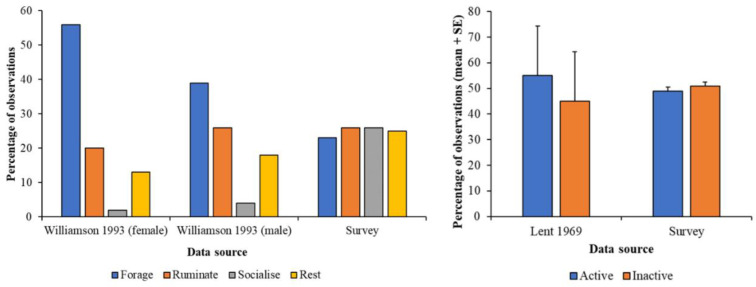
Comparative activity budgets of survey data (records of behaviour noted in each herd) compared to (i) published behaviour of lechwe herds by Williamson 1993 (**left**) and (ii) number of animals inactive or active compared to Lent 1969 (**right**).

**Table 1 animals-10-01874-t001:** Summary of the population ecology of red and Nile lechwe [10,11,25,26,27].

Species	Subspecies	Range	Habitat Preference	Home Range Size
Southern lechwe	Red lechwe *(K. l. leche)*	Angola Botswana DR of Congo Namibia Zambia Zimbabwe	Floodplain grasslands, shallow water meadows (around permanently inundated swamps) and light woodland and grasslands around the edge of wetlands.	No data
	Kafue Flats lechwe *(K. l. kafuensis)*	Zambia (middle Kafue River system)	Floodplain grasslands, shallow water meadows and light woodland and grassland around wetlands.	Up to 1000 animals per km^2^ 3.5 km range diameter (9.6 km^2^)
	Black lechwe *(K. l. smithemani)*	Zambia (Bangweulu in the north)	As above	No data
	Upemba lechwe *(K. l. anselli)*	DR of Congo (Kamalondo Depression, Upemba wetlands)	Flooded grasslands and wetland margins, light woodland around wetlands.	No data
Nile lechwe	None	Ethiopia South Sudan (Sudd and Machar-Gambella wetlands)	Seasonally flooded grasslands and swamps, around the periphery of deeper swamps.	0.06 animals per km^2^ in the dry season.

**Table 2 animals-10-01874-t002:** Population details for the zoos that were sampled (correct as of 31 December 2018).

Region	Years Held	Lechwe & Number of Herds	Population (Male:Female:Unknown)	No. of Calves (<12 Months Old)	Population Age Range
AZA	1	Nile (1)	3:0:0	0	>1–5 years old
AZA	18	Nile (1)	6:9:0	1	<1 year to 20 years
AZA	40	Nile (1)	17:20:0	12	<1 year to 15 years
		Red (1)	6:22:0	10	<1 year to 15 years
EAZA	20	Nile (1)	1:5:0	1	<1 year to 20 years
		Kafue Flats (1)	1:8:0	2	<1 year to 10 years
EAZA	3	Nile (1)	4:5:0	3	<1 year to 10 years
EAZA	15	Nile (1)	4:0:0	0	>1–5 years old
		Kafue Flats (1)	2:5:0	0	>1 year to 15 years
EAZA	17	Nile (2)	4:8:0	3	<1 year to 20 years
			4:0:0	0	6–15 years old
EAZA	3	Nile (1)	1:3:0	0	>1–5 years old
EAZA	45	Nile (1)	0:11:0	0	>1 year to 20 years
	43	Kafue Flats (1)	0:35:0	0	>1 year to 20 years
EAZA	5	Nile (1)	1:4:0	0	>1 year to 10 years
EAZA	11	Nile (1)	1:3:0	0	>1 year to 15 years
EAZA	3	Nile (1)	1:7:0	0	>1–5 years old
EAZA	13	Nile (1)	3:9:0	3	<1 year to 15 years
	15	Kafue Flats (1)	1:12:0	0	>1 year to 15 years
EAZA	30	Nile (1)	3:7:0	0	>1 year to 15 years
		Kafue Flats (1)	35:70:0	12	<1 year to >20 years
EAZA	39	Nile (1)	0:3:0	0	11–15 years old
EAZA	40	Nile (1)	2:5:0	2	<1 year to 5 years
EAZA	No data	Red (1)	2:20:0	0	6–20 years old
EAZA	14	Kafue Flats (1)	9:16:0	7	<1 year to 15 years
EAZA	45	Kafue Flats (1)	2:9:0	0	>1–15 years old
EAZA	10	Kafue Flats (1)	1:5:0	0	>1–15 years old
EAZA	29	Kafue Flats (1)	1:6:0	0	>1–10 years old
EAZA	12	Kafue Flats (1)	1:2:0	0	6–15 years old
EAZA	45	Kafue Flats (1)	2:9:0	1	<1 year to 15 years
EAZA	8	Kafue Flats (1)	2:24:0	No data	No data
EAZA	13	Red (1)	0:23:0	0	>1–10 years old
EAZA	13	Nile (1)	2:5:0	0	6–15 years old

**Table 3 animals-10-01874-t003:** Enclosure sizes and wetland features provided to the lechwe in the sample population.

Region	Species	Number Held	Intended Future Population	House (m^2^)	Paddock (m^2^)	Hardstand (m^2^)	Wetland Provided?/Active Wetland Management?	Wetland (m^2^)
EAZA	Kafue	9	11	216	5670	216	Yes/No	No data
EAZA	Nile	6	11	216	5670	216	Yes/No	No data
EAZA	Nile	9	10	54	5000	27	Yes/No	5000
EAZA	Kafue	7	11	27	38,767	14,038	Yes/No	No data
EAZA	Nile	4	No data	27	38,767	14,038	Yes/No	No data
EAZA	Nile	12	No data	24	30,000	2070	Yes/No	300
EAZA	Nile	4	8	No house	50,000	3000	Yes/No	300
EAZA	Nile	4	6	No house	15,000	None	No wetland	None
EAZA	Kafue	35	16	94	150,000	94	Yes/No	1000
EAZA	Nile	11	9	70	13,500	70	No wetland	None
AZA	Nile	3	3	46	50	3480	Yes/No	37
EAZA	Nile	5	9	30	11,400	120	Yes/No	300
AZA	Nile	15	20	70	16,000	3000	Yes/No	3000
EAZA	Nile	4	No data	55	2500	None	No wetland	None
EAZA	Nile	8	8	120	7500	120	No wetland	None
EAZA	Kafue	13	0	25	16,500	108	Yes/No	50
EAZA	Nile	12	13	47	1300	108	No wetland	None
AZA	Nile	37	29	1486	194,249	None	Yes/No	8000
AZA	Red	28	No data	84	20,234	None	Yes/No	446
EAZA	Kafue	105	No data	No house	138,000	None	Yes/No	12,426
EAZA	Nile	10	No data	No house	29,149	None	Yes/No	6348
EAZA	Nile	3	0	150	16,187	170	Yes/No	70
EAZA	Nile	7	10	50	100	1000	Yes/No	No data
EAZA	Red	22	22	24	101,100	72	Yes/Yes	232
EAZA	Kafue	25	18	60	2940	None	No wetland	None
EAZA	Kafue	11	No data	230	2000	300	No wetland	None
EAZA	Kafue	5	5	80	800	20	No wetland	None
EAZA	Kafue	7	10	16	5000	None	No wetland	None
EAZA	Kafue	3	4	14	4000	None	Yes/No	20
EAZA	Kafue	11	15	34	5490	35	Yes/No data	No data
EAZA	Kafue	26	26	48	24,300	None	Yes/No	160
EAZA	Red	23	23	24	36,422	None	Yes/No	4046
EAZA	Nile	7	7	38	8190	75	No wetland	None

**Table 4 animals-10-01874-t004:** Predictors of lechwe paddock size from responses provided to the survey.

Variable	Estimate	Standard Error (SE)	T Value	*p* Value	Q Value
Species	−15,044.8	12,920.8	−1.164	0.258	0.03
Years kept	479.52	417.11	1.150	0.263	0.035
Region	−16,055.24	17,146.7	−0.936	0.360	0.045
Total in herd	4269.1	1151.7	3.707	0.0012	*0.005 **
Wetland present (yes/no)	4.253	3.358	1.267	0.218	0.025
Intended herd size	−3398.0	1138.4	−2.985	0.007	*0.01 **
Drive-through enclosure (yes/no)	13,917.1	12,566.1	1.108	0.280	0.04
How open is the paddock (%)	531.6	286.3	1.857	0.08	0.015
Mixed-species exhibit (yes/no)	−6403.4	17,581.4	−0.364	0.720	0.05
Active breeding of the lechwe	−22,495.7	12,673.0	−1.775	0.09	0.02

**Table 5 animals-10-01874-t005:** Binomial GLM with logistic link output for predicting whether lechwe are seen resting or ruminating in wetland areas of their enclosure.

Coefficient	Estimate (±SE)	Z Value	*p* Value
Species	Rumination −0.521 (1.66) Resting −0.0703 (1.43)	−0.314 −0.049	0.753 0.961
Wetland (medium)	Rumination −0.606 (2.05) Resting 0.371 (1.70)	−0.296 0.219	0.767 0.827
Wetland (small)	Rumination −2.63 (2.35) Resting −1.02 (1.74)	−1.12 −0.586	0.264 0.558
Wetland (unknown)	Rumination −21.6 (4168.34) Resting −19.90 (4291.41)	−0.005 −0.005	0.996 0.996
Herd size	Rumination −0.013 (0.076) Resting −0.0082 (0.061)	−0.176 −0.134	0.860 0.894
Paddock size	Rumination 0.000015 (0.000036) Resting 0.000019 (0.000029)	0.420 0.656	0.674 0.512
Space per animal	Rumination −0.00098 (0.0011) Resting −0.0007 (0.00082)	−0.894 −0.904	0.371 0.366

**Table 6 animals-10-01874-t006:** Binomial GLM output for predictors of stereotypic behaviour performed from all returned surveys.

Variable	Estimate (±SE)	Z Value	*p* Value
Browse provided (yes/no)	0.31 (2.1)	0.146	0.884
Mixed-species enclosure	−2.95 (2.5)	−1.191	0.234
Number of feeds per day	−1.02 (1.73)	−0.617	0.537
How open is the paddock (%)	−0.008 (0.04)	−0.214	0.831
Drive-through enclosure (yes/no)	−0.47 (1.71)	−0.278	0.781
Wetland present	0.001 (0.001)	1.109	0.267
Size of house (m^2^)	0.004 (0.012)	0.326	0.744
Size of paddock (m^2^)	−0.0001 (0.0001)	−0.647	0.518
Size of hardstanding (m^2^)	0.0005 (0.001)	0.790	0.430
Intended size of herd	0.33 (0.283)	1.159	0.246
Total number of females in herd	0.374 (0.442)	0.844	0.398
Total herd size	−0.353 (0.405)	−0.872	0.383
Species of lechwe	−3.85 (3.02)	−1.276	0.202

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
