# Peer review of "Specialised for the Swamp, Catered for in Captivity? A Cross-Institutional Evaluation of Captive Husbandry for Two Species of Lechwe"

_animals, 2020, doi:10.3390/ani10101874_

Round 1

Reviewer 1 Report

I congratulate the authors on a robust and relevant investigation of captive lechwe enclosures, features, diets and husbandry regimes across many EAZA and AZA facilities. Please find attached a review pdf with markup and comments for suggestions to improve the manuscript. Hopefully the suggestions are relevant and useful to the authors.

Author Response

Thank you for your kind words on the manuscript. Due to how the edits have been provided, embedded into the PDF file, we have been able to lift them out and copy them to show a point by point list of corrections but all corrections and edits have been accepted and acted upon and these are found as track changes in red in the final version. The only point that we did not action was to change concentrate pellets to concentrated pellets as concentrate is a technical term denoting formulated diet used for zoo herbivores.

Reviewer 2 Report

The article presents the results of a survey that the authors made to zoos housing lechwe. The survey included questions related to the enclosures, feeding routines and behaviours, among others. The article collects the results of the zoos that answered the surveys and states the importance of evidence-based husbandry to be used in the development of future best practice guidelines for these antelope.  I wonder if these results are not more fitted for indeed developing best practice guidelines for housing these species in captivity and not so much for a research article. The discussion provides very interesting points to take into account for the welfare of these animals in captivity, but it feels somehow disconnected from the results of the survey, which I feel do not support the conclusions that the authors reached. In my opinion, the conclusions should report the results of the study, not make speculations (even if I agree with the authors with their statements related to animal welfare). I think a different approach presenting the welfare needs for these animals and linking them with the current situation reported from the surveys would have made the article more interesting and would have a better flow. However, I am not sure if this is possible with the current information collected with the surveys, as for example, the behaviours displayed by the lechwe do not seem to be properly assessed.

Specific comments follow:

L12:  which particular social organisation do these animals have? If there is not enough space to add more words, I would rephrase this sentence saying that are social animals or that live in herds, but the way it is written now lets the reader wondering what is particular about the social organisation in this species.

L24-26: I do not agree that this research confirms that zoo-based management should reflect the wild ecology of captive species. Even if I agree with your statement, I do not think that with your research you have reached this conclusion, even if you have discussed this issue.’

L28 and 36: the terms ‘lekking’ and ‘lek’ might be unfamiliar for some readers and need to be defined.

L83-84: how does the production of male calves negatively influence the future survival of the breeding female? There is a reference, but would be interesting to add the explanation in the text.

L98: describe the acronyms the first time they appear in the text (EAZA and AZA), since not all readers are familiar with the name of zoological organisations.

L143: what does ‘model name’ mean in this context?

L146: regarding the region that the zoo was in (EAZA or AZA) – was this a binary option? Perhaps should say association instead of region if it was not more specific?

L158: I would write percentage using words if there are no numbers next to the symbol %.

L160: the published guidelines for antelopes? Or which one? It is mentioned previously on the text that there are no published husbandry guidelines apart from the one in 1993 (L100-101). This gets confusing during the reading of the text; it is not clear to which guidelines the authors sometimes refer to. I would write the full name every time to avoid confusion or having the reader to check continuously the reference list.

L166-199: I have a lot of issues with the behaviour information. It is not clear how the zoos answered this part of the survey. How often did the observations take place? Where the caretakers the observers? If they saw the animal ruminating once, that counted as ‘rumination behaviour occurs in this zoo’? How were the behaviours to be observed chosen? It is not clear which behaviours were actually observed and they keep appearing suddenly on the text: first, social-forage-rest-rumination; then, stereotypic behaviour; then, aggressive behaviour… Which type of stereotypic behaviours were observed in the different zoos? A list of the behaviours observed needs to be provided at the beginning of this section.

Results section: text should be presented before the tables and the tables should be mentioned in the text. That it is not the case for tables 2, 3 and 5.

Table 2: what does the * mean? Consistency in writing yrs or yrs old, one of the two options should be chosen.

L210: the survey results suggested that… but the real number can be found in Species360? I would perhaps write this different. I do not think that the word ‘suggested’ is used correctly here.

Table 4: what dies the * mean?

Figure 2: not sure if this figure adds interesting information that cannot be summarised in the text.

L241-244: this text should be presented before the table. Interspecific aggression observations should be added to the list of behaviours that were asked to be reported.

Figure 4: I think information about how the self-reported observations were done is missing in materials and methods.

L310: which stereotypic/abnormal behaviours were reported?

Discussion and conclusion: I found it very interesting to read, but some conclusions are not supported by the results. For example, in L 351-353 ‘Due to the simplicity of its collection, behavioural data gathered by this survey may be in lacking in its abilities to pinpoint the overall importance of wetland areas to lechwe physical and psychological health.’ But later on (L 427-429) this statement is written ‘Zoos that do not provide wetland are depriving individual lechwe of a key ecological requirement and this should be considered in light of animal welfare needs and species-appropriate husbandry’. Even if I agree with your statement, I think that the results of the survey and especially of the behaviour reports do not allow to reach certain conclusions or make certain statements written in the discussion and conclusions sections.

L468: what does ARB mean? Abnormal repetitive behaviours?

Author Response

Replies to reviewer 2

The article presents the results of a survey that the authors made to zoos housing lechwe. The survey included questions related to the enclosures, feeding routines and behaviours, among others. The article collects the results of the zoos that answered the surveys and states the importance of evidence-based husbandry to be used in the development of future best practice guidelines for these antelope.  

I wonder if these results are not more fitted for indeed developing best practice guidelines for housing these species in captivity and not so much for a research article.

Thank you for your comment. We feel that there is the basis for providing evidence for best practice guidelines from this research, but as we were attempting to see what zoos do with their lechwe as well as determine where there is any standardised practice, we have shown the need for more refined study to be undertaken that would build a best practice approach. So given that this is exploratory and investigative, we feel it is a research paper.

The discussion provides very interesting points to take into account for the welfare of these animals in captivity, but it feels somehow disconnected from the results of the survey, which I feel do not support the conclusions that the authors reached.

We have attempted to re-draft the conclusion and we have removed some of the more superfluous information from the discussion and restructured some sections to improve clarity.

In my opinion, the conclusions should report the results of the study, not make speculations (even if I agree with the authors with their statements related to animal welfare). I think a different approach presenting the welfare needs for these animals and linking them with the current situation reported from the surveys would have made the article more interesting and would have a better flow. However, I am not sure if this is possible with the current information collected with the surveys, as for example, the behaviours displayed by the lechwe do not seem to be properly assessed.

We have moved some of the more discursive elements of the conclusion into the discussion and we have stated the overall findings in the conclusion more clearly. We have evaluated in the discussion that the measure of animal behaviour recording is crude, and that further study should be undertaken. We have presented results that show what lechwe are reported to do and where, to provide zoos with information that could help improve enclosure design.

Specific comments follow:

L12:  which particular social organisation do these animals have? If there is not enough space to add more words, I would rephrase this sentence saying that are social animals or that live in herds, but the way it is written now lets the reader wondering what is particular about the social organisation in this species.

Thank you for the comment. We have edited accordingly.

L24-26: I do not agree that this research confirms that zoo-based management should reflect the wild ecology of captive species. Even if I agree with your statement, I do not think that with your research you have reached this conclusion, even if you have discussed this issue.’

We have edited this part of the abstract.

L28 and 36: the terms ‘lekking’ and ‘lek’ might be unfamiliar for some readers and need to be defined.

We have added in a definition and a reference.

L83-84: how does the production of male calves negatively influence the future survival of the breeding female? There is a reference, but would be interesting to add the explanation in the text.

We have provided a brief explanation.

L98: describe the acronyms the first time they appear in the text (EAZA and AZA), since not all readers are familiar with the name of zoological organisations.

We have edited this in.

L143: what does ‘model name’ mean in this context?

The name that we gave to the model in R during statistical analysis.

L146: regarding the region that the zoo was in (EAZA or AZA) – was this a binary option? Perhaps should say association instead of region if it was not more specific?

Edited.

L158: I would write percentage using words if there are no numbers next to the symbol %.

Edited.

L160: the published guidelines for antelopes? Or which one? It is mentioned previously on the text that there are no published husbandry guidelines apart from the one in 1993 (L100-101). This gets confusing during the reading of the text; it is not clear to which guidelines the authors sometimes refer to. I would write the full name every time to avoid confusion or having the reader to check continuously the reference list.

Edited to avoid confusion.

L166-199: I have a lot of issues with the behaviour information. It is not clear how the zoos answered this part of the survey. How often did the observations take place? Where the caretakers the observers? If they saw the animal ruminating once, that counted as ‘rumination behaviour occurs in this zoo’? How were the behaviours to be observed chosen? It is not clear which behaviours were actually observed and they keep appearing suddenly on the text: first, social-forage-rest-rumination; then, stereotypic behaviour; then, aggressive behaviour… Which type of stereotypic behaviours were observed in the different zoos? A list of the behaviours observed needs to be provided at the beginning of this section.

We have edited this to improve clarity.

Results section: text should be presented before the tables and the tables should be mentioned in the text. That it is not the case for tables 2, 3 and 5.

Table 2 is mentioned on page 4, before the table.

We have edited for Tables 3 and 5.

Table 2: what does the * mean? Consistency in writing yrs or yrs old, one of the two options should be chosen.

Edited

L210: the survey results suggested that… but the real number can be found in Species360? I would perhaps write this different. I do not think that the word ‘suggested’ is used correctly here.

Edited

Table 4: what dies the * mean?

Edited in the text

Figure 2: not sure if this figure adds interesting information that cannot be summarised in the text.

We have deleted this figure.

L241-244: this text should be presented before the table. Interspecific aggression observations should be added to the list of behaviours that were asked to be reported.

Edited and provided detail.

Figure 4: I think information about how the self-reported observations were done is missing in materials and methods.

We have added in this detail.

L310: which stereotypic/abnormal behaviours were reported?

We have added in this detail

Discussion and conclusion: I found it very interesting to read, but some conclusions are not supported by the results.

We are pleased that you found the discussion interesting. We have amended the conclusions to make them more specific to the key findings of the paper.

For example, in L 351-353 ‘Due to the simplicity of its collection, behavioural data gathered by this survey may be in lacking in its abilities to pinpoint the overall importance of wetland areas to lechwe physical and psychological health.’ But later on (L 427-429) this statement is written ‘Zoos that do not provide wetland are depriving individual lechwe of a key ecological requirement and this should be considered in light of animal welfare needs and species-appropriate husbandry’. Even if I agree with your statement, I think that the results of the survey and especially of the behaviour reports do not allow to reach certain conclusions or make certain statements written in the discussion and conclusions sections.

We have amended this points but we feel that it is important to keep in, based on the positive comments from other reviewers too. We do not link these two sentences. We state that the behavioural information from the surveys is quite crude (collected on the go by busy zoo keepers) but we do present important information from the literature that these specialised aquatic antelope spend a great deal of time in the water and therefore zoos need to consider this information in light of best practice approaches. We have edited the statement to make it less “pointed” and we have suggested a way of informing practice further with extra research.

L468: what does ARB mean? Abnormal repetitive behaviours?

We have ensured that ARB is included as the abbreviation the first time the term is mentioned.

Reviewer 3 Report

See attached file.

Author Response

Replies to reviewer 3

Specialised for the swamp, catered for in captivity? A cross-institutional evaluation of captive husbandry for two species of lechwe The following paper examined two species of zoo-housed lechwe, which focused on survey responses from 26 zoos about 33 groups/herds. Details included enclosure design, herd makeup, diet, enrichment, and abnormal behaviors, with a focus on using this information to assess current housing conditions and guide future management. Below, I have combined specific edits and concerns, as well as suggestions for future revisions:

Lines 12-13 (Simple Summary): “…have a particular social organisation” should be rephrased to say what is the social organization, similar to what is stated in the Absract.

Edited.

Lines 48-51: Both these sentences require citations to support these claims. A few similar studies have been conducted or proposed with ungulates, including Tennant et al. (2018) with hippos in North American zoos. Similarly, Fernandez, Yoakum, and Andrews (2020) examined the daily and annual activity of two zoo-housed grizzly bears, with attention to their wild counterparts. Other papers have discussed the issue, including Fernandez and Timberlake (2008) with respect to zoo-academic collaborations on behavior- and conservation-related research, as well as Veasey, Waran, and Young (1996) and Hutchins (2006) in terms of using the behavior of a species’ wild counterparts to guide zoo management. The latter article (Hutchins, 2006) discusses some of the difficulties and problems with using this metric for elephants, which is equally applicable to other species. Regardless, it would be good to more specifically address and cite these claims, include the above references, and additionally cite other survey-based research, like Tennant et al., 2018, that examine the housing arrangements of zoo animals and/or compare the activity of zoo animals to the same wild species. Overall, I think this is what the Introduction is lacking; more detail on why the reader should care about both surveying how some species of zoo animals are housed, and what benefits are gained from comparing all things zoo-housed to the same or similar wild populations. The authors spend considerable detail in the Introduction focused on lechwe wild populations and the importance of attending to such factors when housing lechwe. Now the authors just need to inform the reader how using survey research such as this can better help assess all the factors related to the housing of any captive species, and equally important, why we bother comparing zoo animals to wild animals.

Expanding the first paragraph (lines 48-57) into 2-3 paragraphs addressing both these issues would greatly facilitate the discussion and purpose of this study.

We have included more details and useful references around the points that you have noted and we have included some of the ungulate specific references that you kindly provided. We hope that this further sets the scene for why the research is being conducted.

Lines 53-54: Need a reference for the claim about the 863 Species360 zoos, which will also provide a reference for the NGO that is “Species360”.

Edited

Lines 59-60: Remove “contentious” from the sentence. Something like, “Lechwe taxonomy is contested between two or three different species, including…”

Edited

Materials and Methods: More information about the surveys itself should be provided. I am assuming that most of the questions were open-ended, but that needs to be clearly stated, as well as if it is true for all questions. It would be helpful to provide sample questions asked, as well as who provided answers to those survey questions (Curators? Keepers? Did they have to work for the zoo?) and how responses were delivered to the authors.

We have included all of these details; example questions and how the survey was distributed as well as who was the required recipient. We will gladly include the survey as an appendix if this would be helpful.

Data Analysis: Need to provide more information initially about tests done to establish normality and homogeneity of variance, and how such data were then treated. Some of this information is provided in the following sub-sections, which is fine, but initially, I would first address this issue here, as I assume this sets up why some of the tests conducted were used.

We have included some preamble to the statistical analysis section to outline key points. Anderson-Darling tests were used for each specific subset of data from the main sample to check for normality. We did not do any transformations of data if these data were not normal. Model fits were checked in R Studio, based on plots of the residuals for the model and r2 values were used (and stated) to assess variation captured by the model). We do not wish to move all of the normality testing to one paragraph at the start as we have clearly and thoroughly explained how each specific section of data have been dealt with so the reader can see how each question has been analysed and how that analysis has been chosen.

Line 143: “anova(model name)” function should be put in quotes.

Edited

Line 212: Missing “lechwe” after “southern”.

Edited

Results: For Table 2 & 3, it does not make much sense to provide the specific details of each institution. It would make more sense to summarize these details in terms of EAZA versus AZA zoos and with descriptive statistics of the range, mean, median, and/or some measure of variance for each. The authors could even group these as categories (greater than ‘x’) if they wanted to emphasize binomial distrubtions around herd or enclosure size (i.e., herds/wetland size is either big or small).

We originally considered this but based on the small sample of AZA zoos we did not feel that this was appropriate, and given that there are likely to be readers who are interested in individual responses as well as seeing the degree of variation in herd size and structure, as well as the extreme variation in housing sizes we wish to keep these tables as they are. They provide useful information on the results from survey and all the reader to see the types of questions that we asked.

Line 219-221: Spearman’s rank correlations are typically represented with “rs”. “r” is usually reserved for Pearson’s correlations.

Edited

Table 5: I am not sure it is useful to provide a table with all of this information. I think it would be simpler to state that all lechwe, when housed in a mixed-species exhibit, included either other ungulates or birds, and then give some examples of the species. Other descriptive statistics could be provided if desired, including the range (1-8) of the lechwe and other species included in such mixed species exhibits.

Edited as per your suggestion. We have suggested this table be moved to a supplementary file.

Figure 3: This could be demonstrated more clearly by eliminating each zoo’s individual responses, and then providing one or two graphs that showed the differences between the guidelines and observed percentages. Significant differences should be labeled with such a graph (I would use lines between the bars), and the measure of variance used should be stated (standard error of the mean or standard deviation?)

We have re-drawn this figure to show the survey average and the published guidelines only. Thank you for the suggestion.

Figure 4: I am not sure a graph is necessary here, as this is just the amount of zoos that reported that their animals engaged in these behaviors. It is likely more idealized than a fair representation of occurrence, and the details in the graph could be provided by simply stating the range of occurrence zoos reported for these key-state behaviors across the different enclosure zones.

We appreciate that this graph is crude, but we feel it does show visually where lechwe perform behaviours that are important for their welfare and hence could provide useful information to build on for future research into what exactly is the importance of each zone for the behaviours noted. We would like to keep it in as it shows how we have used the responses from the surveys.

Table 6 & 7 and Figure 5 are three of the best summaries of the results of these surveys. Excellent job.

Thank you.

Lines 329-334: Really long sentence. Break up into two separate sentences.

Attempted to edit. Already edited based on other reviewer comments but have tried to rationalise.

Line 335: Replace “too” with “as well” or the like.

Edited

Line 352: Remove “in” before “lacking”.

Edited

Line 356: Worth citing Brereton (2020) with respect to measuring space utilization, and could also potentially benefit from citations of the first author’s work on measuring enclosure use.

We have included information on enclosure usage in practice as well as two other papers that are original methodological papers that explain how to measure space use. We further expanded on the limitations of our behavioural data and how it could be expanded with these other measures.

Lines 449-451: I think it needs “on” between “based” and ‘wetland”.

Edited

Line 468: Remove “And” at the start of the sentence.

Edited

Line 468-469: If you are going to use an abbreviation here for ‘abnormal repetitive behavior’ (ARB), it should be spelled out. However, I think this is the first time this abbreviation is introduced, so I would either introduce the acronym in the Introduction, or eliminate its usage altogether.

We have edited this based on previous reviewer comments so this should now be rectified.

Line 485: Re-phrase, as this starts the Conclusions off a bit more informally. Overall, the Discussion reads fine, but I think some work could be done to (a) reduce the length, with a focus on minimizing a detailed walkthrough of the results, and (b) emphasizing that not much can be said behaviorally, since the extent of this survey simply asked one or more persons at each institution to self-report whether their animals engaged in these activities. As noted above, more detail needs to be provided for who answered these questions and what those questions were, but regardless, this is a notoriously bad way to determine anything about the behavior of any animal in any context. It is understandable, as the authors could not expect any of these facilities answering such a survey to conduct their own activity budget-based study. Still, it is a considerable limitation on any claims about about the general activity of the lechwe other than that most of these behaviors appeared to occur and the very general amount of time spent being active verse inactive matched those reported by Lent (1969). So, it would be worth being clear that this was a limitation (beyond the only mention of this in line 351-353), and then greater caution should be taken in suggesting that any of the behaviors observed, including the overall use of the different types of enclosure areas, represent some “true” assessment of those behavioral distributions.

We have edited the discussion and re-written the conclusion based on the comments of the first two reviewers, plus these suggests too. We have attempted to expand on the limited information gleaned on lechwe activity from the survey, but we still feel that it is relevant to our understanding of animal welfare and lechwe management to have this information on where behaviour is observed, presented in this paper and evaluated.  

We have attempted to include more information on what we have discovered about lechwe from our results and have integrated this into the discussion.

Aside from some of these criticisms, this is a good and necessary study. We need more information about how animals are housed in zoos and/or other captive settings and how that compares to their wild counterparts. The authors have done a commendable job in providing a resource that will hopefully help dictate future management of lechwe. Also, by editing their Introduction to outline both why zoo/wild comparisons are important and how surveys of current management practices can aide such assessments, the paper can serve a bigger purpose in the literature. Namely, to emphasize the future of zoos as empirical institutions.

Thank you for the positive comments and helpful review suggestions.

Round 2

Reviewer 2 Report

I am pleased with the new version that the authors presented, and I think that the quality of the manuscript has substantially improved. My main concern was that the conclusions and part of the discussion were not supported by the results, but I feel that the current version has corrected that. I only have small comments related with typos and style form:

Introduction, discussion, and conclusions: text needs to be justified.

L17: there are two periods and an extra space.

L167: there is an extra space between resting/lying down.

L171: maybe you wanted to say ‘stereotypic behaviour’ instead of ‘’stereotypic behaviour?

L196: the subheading should be one line below.

L269: there is an extra s in institution.

L305: there is an extra 9 at the end of the sentence.

L425: there are two commas and extra spaces after Table 3.

L548: a period is needed instead of a comma in front of We.

L602: an extra space is needed after validity,.

Author Response

I am pleased with the new version that the authors presented, and I think that the quality of the manuscript has substantially improved. My main concern was that the conclusions and part of the discussion were not supported by the results, but I feel that the current version has corrected that. I only have small comments related with typos and style form:

Thank you for the comments. We are pleased that the new manuscript is improved and has answered all of your comments and concerns.

Introduction, discussion, and conclusions: text needs to be justified.

Edited.

L17: there are two periods and an extra space.

Edited 

L167: there is an extra space between resting/lying down.

Edited

L171: maybe you wanted to say ‘stereotypic behaviour’ instead of ‘’stereotypic behaviour?

Edited.

L196: the subheading should be one line below.

Edited.

L269: there is an extra s in institution.

Edited (not we also corrected "animal" to "animals" in this sentence too).

L305: there is an extra 9 at the end of the sentence.

Edited.

L425: there are two commas and extra spaces after Table 3.

Edited.

L548: a period is needed instead of a comma in front of We.

Edited.

L602: an extra space is needed after validity,.

Edited.